# ScnR1-Mediated Competitive DNA Binding and Feedback Inhibition Regulate Guvermectin Biosynthesis in *Streptomyces caniferus*

**DOI:** 10.3390/biology14070813

**Published:** 2025-07-04

**Authors:** Haoran Shi, Jiabin Wang, Xuedong Zhang, Na Zhou, Xiangjing Wang, Wensheng Xiang, Shanshan Li, Yanyan Zhang

**Affiliations:** 1State Key Laboratory for Biology of Plant Diseases and Insect Pests, Institute of Plant Protection, Chinese Academy of Agricultural Sciences, No. 2 Yuanmingyuan West Road, Haidian District, Beijing 100193, China; shr05192023@163.com (H.S.); jbwang0725@163.com (J.W.); zhangxuedong1206@163.com (X.Z.); 15826636007@163.com (N.Z.); xiangwensheng@neau.edu.cn (W.X.); 2Key Laboratory of Agricultural Microbiology of Heilongjiang Province, Northeast Agricultural University, No. 59 Mucai Street, Xiangfang District, Harbin 150030, China; wangneau2013@163.com

**Keywords:** guvermectin, *Streptomyces*, ScnR1, transcriptional regulation, secondary metabolism

## Abstract

Guvermectin, a *Streptomyces*-derived purine nucleoside, exhibits potent plant growth-promoting and broad-spectrum antibacterial activities, making it a promising agent for sustainable agriculture. Identification and characterization of novel transcriptional regulators involved in guvermectin biosynthesis is crucial for production improvement via transcriptional regulator engineering. This study identified ScnR1, an extra-cluster regulator influencing guvermectin production. The regulatory role of ScnR1 in guvermectin biosynthesis and its complex interactions with GvmR and GvmR2 were investigated. These findings advance our understanding of the multi-layered regulation of secondary metabolism in *Streptomyces*, offering a theoretical foundation for enhancing natural product yields through transcriptional regulator engineering.

## 1. Introduction

Guvermectin, also referred to as decoyinine or angustmycin A, is a purine nucleoside natural product (NP) derived from *Streptomyces*, featuring an unusual psicofuranose linked to adenine via a distinctive C−N glycosidic bond [1,2,3,4]. This NP exhibits significant plant growth-promoting and yield-enhancing properties [5]. Field trials have demonstrated that seed soaking with guvermectin leads to substantial increases in rice yield, with improvement ranging from 6.2% to 19.6%, outperforming other commonly used plant growth regulators such as cytokinins, brassinosteroids, and a gibberellin-auxin-brassinosteroid mixture [5]. Owing to its exceptional impacts on rice growth and yield, guvermectin was officially registered as a new biochemical pesticide in China in 2021 (registration number: PD20212929). Additionally, guvermectin demonstrates potent inhibitory efficacy against multiple plant pathogenic bacteria, including *Xanthomonas oryzae* pv. *Oryzae* (causal agent of rice bacterial leaf blight), *Ralstonia solanacearum* (tomato bacterial wilt pathogen), and *Pseudomonas syringae* (lilac bacterial blight) [6], highlighting its potential as a broad-spectrum biopesticide for disease management in agriculture.

Several *Streptomyces* species, including *Streptomyces angustmyceticus* NBRC 3934, *Streptomyces decoyicus* NRRL 2666, and *Streptomyces caniferus* NEAU6, possess the ability to produce guvermectin [5]. The BGCs responsible for guvermectin production in these strains share high sequence similarity [3]. Each BGC consists of nine genes: six encoding essential biosynthetic enzymes (*gvmA*, *gvmB*, *gvmC*, *gvmD*, *gvmE*, and *gvmF*), one transcriptional regulatory gene (*gvmR*), and two major facilitator superfamily (MFS) transporter-encoding genes (*gvmT1* and *gvmT2*). Previous studies employing heterologous expression, genetic manipulation, and biochemical characterization have elucidated the guvermectin biosynthetic pathway, which involves a series of enzymatic reactions, with D-fructose 6-phosphate and ATP as precursors to synthesize guvermectin [3,7]. *gvmR*, the only cluster-situated regulator (CSR) in the guvermectin BGC, encodes a LacI-family regulator. GvmR is crucial for guvermectin biosynthesis, as it activates the expression of the guvermectin BGC through direct binding to the *gvmR*, *gvmA*, and *O1* promoters [8]. In addition, genomic analysis has identified an additional LacI-family regulator gene, *gvmR2*, situated around 8 kb upstream of the guvermectin BGC (relative to *gvmT1* position). GvmR2 has been shown to repress the guvermectin BGC expression through competitive binding with GvmR at the promoter regions of *gvmR*, *gvmA*, and *O1* [8]. However, to date, aside from the two proximal transcriptional regulators (TRs), GvmR and GvmR2, little is known about the TRs located far from the guvermectin BGC that may influence its biosynthesis. For streptomycete-derived NPs like guvermectin, biosynthesis is typically tightly controlled by a complex, hierarchical network involving cluster-situated regulators and a plethora of pleiotropic/global regulators [9,10,11,12]. This regulatory complexity is one of the key factors contributing to the low production efficiency of these compounds [13,14,15]. Therefore, in-depth exploration of novel TRs involved in guvermectin biosynthesis, along with an investigation into their regulatory mechanisms, will enhance our understanding of the biosynthetic regulatory network and provide a solid theoretical foundation for the construction of engineered strains capable of efficient production of guvermectin.

In this study, we utilized the 14 bp palindromic sequence: 5′-RTCATWCGYATGAY-3′, previously identified as essential for GvmR binding [8], to scan the *S. caniferus* NEAU6 genome using the MAST/MEME web-based application. This analysis led to the identification of a new LacI-family TR gene, *scnR1* (*scn1544*). Two complete 14 bp GvmR binding motifs were found in the promoter region of *scnR1*, and the direct activation of *scnR1* by GvmR was further confirmed through electrophoretic mobility shift assays (EMSAs) and transcriptional analysis. Notably, *scnR1* overexpression was found to inhibit the biosynthesis of guvermectin, with this inhibitory effect mediated through competitive binding with GvmR at the *gvmR*, *gvmA*, and *O1* promoter regions. Additionally, a reciprocal feedback inhibition mechanism between ScnR1 and GvmR2 was uncovered. The competitive binding of ScnR1 with both GvmR and GvmR2 at the same target promoters, along with their mutual feedback regulation, reveals a multi-layered regulatory model governing guvermectin biosynthesis. These findings enrich our understanding of the intricate and coordinated mechanisms that regulate NP biosynthesis in *Streptomyces* species.

## 2. Materials and Methods

### 2.1. Bacterial Strains, Plasmids, Primers, and Growth Conditions

Appendix A list the strains, plasmids, and primers utilized in the present work. *Streptomyces caniferus* NEAU6 is a natural producer known for producing guvermectin [3]. ΔgvmR (a mutant with *gvmR* insertional inactivation), NEAU6/gvmR (a *gvmR* overexpression strain) and NEAU6/P_hrdB_gvmR2 (a *gvmR2* overexpression strain) are three derivatives generated from *S. caniferus* NEAU6 in previous studies [3,8]. *Escherichia coli* JM109 served as the plasmid-propagating host, while plasmid transfer into *Streptomyces* strains was achieved via intergeneric conjugation using the donor strain *E. coli* ET12567/pUZ8002 [16]. Media formulations used for spore collection, vegetative mycelium preparation, and guvermectin biosynthesis in *S. caniferus* NEAU6 and its derived mutants were prepared as previously described [8]. For *E. coli* cultivation, LB medium was used, while MS agar plates were employed to facilitate conjugation between *E. coli* donor cells and *Streptomyces* recipients [16].

### 2.2. Gene Overexpression and Disruption

To achieve overexpression of *scnR1* in *S. caniferus* NEAU6, we constructed the plasmid pSET152::P_hrdB_scnR1. To construct pSET152::P_hrdB_scnR1, the promoter region of *hrdB* and the *scnR1* open reading frame were obtained by PCR-amplification from the genome of *S. coelicolor* M145 and *S. caniferus* NEAU6, respectively, using PhrdB-F/R and HscnR1-F/R primer sets. These DNA segments were subsequently inserted into *Eco*RI/*Bam*HI-digested pSET152, resulting in the pSET152::P_hrdB_scnR1 construct. This construct was then transferred into *S. caniferus* NEAU6 via *E. coli* ET12567/pUZ8002-mediated intergeneric conjugation, producing the overexpression strain NEAU6/P_hrdB_scnR1.

*scnR1* was deleted through double-crossover homologous recombination. Briefly, two flanking regions of *scnR1* (scnR1-L: 2021 bp; scnR1-R: 2043 bp) were amplified using R1LF/R and R1RF/R primers, respectively. The resultant scnR1-L and scnR1-R fragments were directionally inserted into the *Eco*RI/*Hind*III restriction sites of pKC1139, generating *scnR1* deletion plasmid pKC1139::scnR1. Introduction of this recombinant plasmid into *S. caniferus* NEAU6 generated the *scnR1* deletion strain (ΔscnR1), which was subsequently confirmed through PCR amplification using primer pairs veri-F1/R1 and veri-F2/R2, followed by DNA sequencing.

### 2.3. Detection of Guvermectin

Guvermectin was extracted from the whole-cell fermentation broth by mixing the broth (0.4 mL) with ethanol (1.6 mL) at a 1:4 ratio. The mixture was subjected to ultrasonic treatment for one hour, then centrifuged at 12,000× *g* for ten minutes. The supernatant was collected and analyzed via HPLC analysis as described previously [8]. HPLC was performed using an Agilent 1260 II HPLC system equipped with a ZORBAX SB-Aq column (4.6 mm × 250 mm, 5 μm) (Agilent Technologies, Santa Clara, CA, USA). The mobile phase consisted of ultrapure water and HPLC-grade acetonitrile (9:1, *v*/*v*), with a flow rate of 0.8 mL/min. The column temperature was maintained at 28 °C, and detection was carried out at 260 nm. The total run time for each sample was 30 min. The guvermectin standard had a purity of 94%. A linear calibration curve was established over a concentration range of 0.005–0.5 mg/mL, with the following regression equation: Y = 18627X + 87.87, where Y represents the peak area (mAU·min) and X represents the guvermectin concentration (mg/mL). The correlation coefficient (R^2^) was 0.9996.

### 2.4. Isolation of RNA and qRT-PCR Analysis

Total RNAs were isolated from *S. caniferus* NEAU6 and derivatives grown in guvermectin fermentation media at two time points (day 2 and day 5). RNA extraction, quality assessment, quantity measurement, synthesis of cDNA, and subsequent qRT-PCR were conducted following previously established protocols [17].

### 2.5. Overexpression and Purification of the Three LacI-Family Proteins

The ScnR1 overexpression plasmid was generated by PCR amplification of the *scnR1* coding sequence from the *S. caniferus* NEAU6 genome using primers GEXscnR1-F/R. The resulting segment was subsequently ligated into *Bam*HI/*Xho*I-digested pGEX-4T-1, generating *scnR1* expression construct pGEX-4T-1::scnR1. The expression constructs for GvmR and GvmR2, pGEX-4T-1::gvmR, and pGEX-4T-1::gvmR2 had been previously constructed and described in the literature [8]. The overexpression and purification of the proteins were carried out following the procedures outlined in a previous study [18]. The original SDS-PAGE image showing the purity of purified ScnR1 is provided in Appendix A.

### 2.6. EMSAs

Promoter probes of interest were PCR-amplified from *S. caniferus* NEAU6 genome with primer combinations provided in Appendix A. The electrophoretic mobility shift assays (EMSAs) were conducted according to a previously described method with modifications [19]. A total of 20 ng of DNA probe were incubated with varying concentrations of ScnR1, GvmR, or GvmR2 at 25 °C for 25 min in a 20 μL buffer containing 20 mM Tris-HCl (pH 7.5), 1 mM dithiothreitol (DTT), 10 mM MgCl_2_, 0.5 mg/μL calf BSA, and 5% (*v*/*v*) glycerol. After incubation and electrophoresis, the 4% (*w*/*v*) non-denaturing polyacrylamide gels were stained with SYBR Gold Nucleic Acid Gel Stain for 30 min. Gel images were captured using Bio-Rad GelDoc XR (Bio-Rad, Hercules, CA, USA) under ultraviolet transillumination. Original uncropped EMSA images corresponding to the figures of this paper are provided in Appendix A, respectively.

### 2.7. Prediction of ScnR1 Binding Sites Using AlphaFold 3

Predictions of interactions between ScnR1 and its target promoter regions were made using AlphaFold 3 [20]. Structural visualization and analysis of the predicted models were performed using PyMOL 3.1 [21].

### 2.8. Statistical Analysis

All experiments were conducted at least three times. Data are presented as mean values ± standard deviation (n = 3). Statistical significance was demonstrated using Student’s *t*-test (two-tail). *** *p* < 0.001, **** *p* < 0.0001, “ns” indicates not significant.

### 2.9. Bioinformatics Analysis

Pfam 37.4 (https://pfam.xfam.org/, accessed on 2 July 2025), SMART version 9 (http://www.smart.emblheidelberg.de/, accessed on 2 July 2025), and KEGG Release 115.0 (http://www.genome.jp/kegg/, accessed on 2 July 2025) were used for analyzing potential functional domains. Sequence motifs were characterized using the MEME Suite Version 5.5.8 (https://meme-suite.org/meme/index.html, accessed on 2 July 2025) [22].

## 3. Results

### 3.1. scnR1, a Highly Conserved LacI-like Regulator, Is a Direct Target of GvmR

Previous biochemical evidence indicated that a 14 bp palindromic sequence (5′-RTCATWCGYATGAY-3′, where R = G/A, W = A/T, Y = T/C) was necessary for the DNA-binding activity of GvmR [8]. We thus wondered whether there are similar palindromic sequences bound by GvmR in the promoter regions beyond the guvermectin cluster. To address this inquiry, we used the 14 bp palindromic sequence to scan the NEAU6 genome with MAST/MEME (http://meme-suite.org, accessed on 2 July 2025). This search yielded 54 promoter regions containing intact or partial sequences resembling the 14 bp binding motif (Appendix A). These candidate regions were further analyzed through EMSAs with GvmR, revealing specific interactions with only one specific promoter probe P_R1_, which corresponded to the promoter region of *scnR1* (*scn1544*) (Figure 1A,B). P_R1_ contained two 14 bp palindromic sequences with 7 bp separating them (Figure 1A). To evaluate the regulatory impact of GvmR on *scnR1* expression, total RNAs were extracted from the mycelia of NEAU6, NEAU6/pSET152, ΔgvmR, and NEAU6/gvmR strains cultivated for 2 and 5 days. Subsequently, qRT-PCR was carried out to measure the transcriptional levels of *scnR1*. *gvmR* inactivation did not result in a significant alteration in *scnR1* transcription, while overexpression of *gvmR* resulted in a significant increase in *scnR1* transcription at day 5 (Figure 1C), indicating that GvmR positively influences *scnR1* expression.

*scnR1* encodes a LacI-family TR, which has high similarity with other LacI-like family members, such as SCO1642 (68% identity) from *Streptomyces coelicolor*, and SBI_8484 (75% identity) from *Streptomyces bingchenggensis*. Interestingly, ScnR1 and its orthologs are highly conserved among *Streptomyces*; a search of the 268 *Streptomyces* genomes obtained from the NCBI Genome database (as of Aug. 2021) revealed 263 ScnR1 orthologs (query coverage 90% to 100% and percent identity 60% to 100%) in 261 *Streptomyces* species. Nevertheless, despite this conservation, their functional roles remain largely unexplored.

### 3.2. scnR1 Overexpression Inhibits Guvermectin Production by Suppressing Expression of the Guvermectin BGC

Considering ScnR1’s status as a highly conserved TR in *Streptomyces*, the question arises: could ScnR1 influence guvermectin production? To explore this hypothesis, an *scnR1* overexpression strain, NEAU6/P_hrdB_scnR1, was constructed, in which *scnR1* expression was controlled by the *hrdB* promoter in an integrative pSET152. Indeed, after flask fermentation, no guvermectin was produced by NEAU6/P_hrdB_scnR1 in comparison with NEAU6 and NEAU6/pSET152 strains (Figure 2A and Appendix A), indicating that *scnR1* overexpression has a strong repressive effect on guvermectin production. We also constructed an *scnR1* deletion mutant (ΔscnR1), in which a 1053 bp fragment within the *scnR1* ORF, including the entire coding region for the HTH-type DNA-binding motif (amino acids 17–89) and the Peripla_BP_3 domain (amino acids 184–342), essential for LacI function, was deleted via homologous recombination. The creation of the ΔscnR1 mutant was confirmed by PCR and DNA sequencing (Appendix A). Notably, guvermectin production in ΔscnR1 was similar to that of the NEAU6 strain, indicating that the native expression levels of *scnR1* in NEAU6 do not significantly affect guvermectin production. A pronounced repression of guvermectin production was only observed under high-expression conditions, such as in the NEAU6/P_hrdB_scnR1 strain.

To elucidate the repressive mechanism underlying guvermectin biosynthesis resulting from *scnR1* overexpression, we conducted qRT-PCR analysis of several genes (*scnR1*, *gvmR*, *gvmA*, and *gvmE*) to determine the influence of ScnR1 on guvermectin BGC. The expression of *scnR1* in NEAU6/P_hrdB_scnR1 was significantly higher than in NEAU6/pSET152 at both sampling time points, validating effective overexpression of *scnR1* (Figure 2B). Transcripts of key guvermectin BGC genes, *gvmR*, *gvmA*, and *gvmE*, were nearly undetectable in NEAU6/P_hrdB_scnR1 compared to NEAU6/pSET152 (Figure 2B), suggesting that *scnR1* overexpression represses guvermectin production through the transcriptional inhibition of the guvermectin BGC.

### 3.3. ScnR1 Binds to the Bidirectional gvmR–gvmA and O1 Promoters

To identify the direct regulatory targets of ScnR1, an N-terminal GST-tagged ScnR1 was expressed and purified, and its binding to the promoter regions within the guvermectin BGC was assessed using EMSAs (Figure 3A,B). As shown in Figure 3C,D, ScnR1 exhibited DNA-binding activity toward the *gvmR*–*gvmA* bidirectional promoter and the *O1* promoter, indicating that ScnR1 represses guvermectin BGC expression through the direct binding to these two promoter regions. Additionally, EMSA results revealed that ScnR1 can also bind to its own promoter region (Figure 3E), suggesting that ScnR1 functions as an autoregulator.

### 3.4. Identification of ScnR1 Binding Sequences

EMSA results revealed that ScnR1 binds to the P_R-A_, P_O1_, and P_R1_ promoter probes in a concentration-dependent manner (Figure 3). Notably, P_R-A_ and P_O1_ were previously verified as direct targets of GvmR [8], and P_R1_ was confirmed as a direct target of GvmR in this study (Figure 1), suggesting that ScnR1 and GvmR can bind to the same target promoters. This observation prompted us to investigate whether ScnR1 recognizes binding sequences similar to those of GvmR. To explore potential DNA-binding sites for ScnR1, we employed AlphaFold 3 to predict regions where ScnR1 might interact with P_R-A_, P_O1_, and P_R1_. The AlphaFold 3 analysis identified two 18 bp sequences (Sites I and II) within the bidirectional promoter region of *gvmR-gvmA* (Figure 4A). These two sites are separated by an 8 bp spacer. Site I spans from positions −198 to −181 relative to the translation start codon (*tsc*) of *gvmA*, while Site II is located from −172 to −155. In the promoter region of *O1*, two 17 bp sequences (Site III: −177 to −161 and Site IV: −77 to −61 relative to the *O1 tsc*) were identified, separated by an 83 bp distance (Figure 4A). Additionally, within the *scnR1* promoter region, we identified a 17 bp sequence (Site V) and an 18 bp sequence (Site VI), located at positions −89 to −73 and −69 to −52, respectively, relative to the *scnR1 tsc* (Figure 4A). Subsequently, we analyzed the six identified binding sites using the MEME Suite bioinformatics platform, which revealed a conserved 14 bp palindromic sequence, 5′-RTSATACGHATRAY-3′ (R = G/A, S = C/G, H = A/T/C, Y = T/C) (Figure 4B). Notably, this 14 bp motif closely matches the previously reported binding motif of GvmR [8], suggesting that ScnR1 may influence the expression of target genes through competitive binding to the same cis-acting elements as GvmR.

To assess the significance of the 14 bp palindromic sequence for ScnR1 DNA-binding activity, three mutated probes (P_R-A_-M1, P_R-A_-M2, and P_R-A_-M3) generated in a previous study were individually subjected to EMSAs with ScnR1 (Figure 4C) [8]. As shown in Figure 4D, with increasing concentrations of ScnR1 (0.05, 0.1, and 0.2 μmol L^−1^), the electrophoretic mobility and intensity of the protein–DNA complexes formed between the single-site mutant probe (P_R-A_-M1 or P_R-A_-M2) and ScnR1 were basically similar to those observed with the native P_R-A_ probe, with all sets of complexes displaying two distinct bands. However, when the ScnR1 concentration was increased to 0.3 μmol L^−1^, the grayscale ratio of the upper band of the complexes formed by the mutant probe (P_R-A_-M1 or P_R-A_-M2) was significantly reduced compared to that of the complexes formed with the wild-type P_R-A_ (Figure 4D and Appendix A). Additionally, the grayscale ratio of the free P_R-A_-M1 probes increased significantly, while the ratio of the free P_R-A_-M2 probes remained comparable to that of the free P_R-A_ probes (Figure 4D and Appendix A). These results indicated that the mutation of site I weakens the binding of ScnR1 to the target promoter at higher concentrations while site II mutation alters the binding pattern without affecting overall binding efficiency. Notably, the binding affinity of the double-site mutant probe P_R-A_-M3 to ScnR1 was further reduced. As the ScnR1 concentration increased, only two relatively weak DNA–protein complexes bands were observed (Figure 4D), suggesting that the two 14 bp palindromic sequences are crucial for ScnR1’s binding activity.

### 3.5. Reciprocal Transcriptional Repression Between scnR1 and gvmR2

Approximately 8 kb upstream of the guvermectin BGC (relative to the *gvmT1* locus), *gvmR2* encodes a LacI-family TR. GvmR2 has been shown to directly repress guvermectin BGC expression, thereby negatively regulating guvermectin production [8]. Here, ScnR1 was also found to negatively regulate the guvermectin BGC. To explore the potential interaction between *scnR1* and *gvmR2*, we employed EMSA and qRT-PCR analysis. The EMSA results showed that, with increasing concentrations of ScnR1, this protein was able to bind to the *gvmR2* promoter region (P_R2_), forming distinct DNA–protein complexes bands (Figure 5A). Similarly, GvmR2 also bound to the promoter region of *scnR1*, forming clearly identifiable complexes bands (Figure 5A). These findings suggested that ScnR1 and GvmR2 may influence each other’s gene expression through direct interaction. Moreover, the transcription of *gvmR2* was significantly reduced in NEAU6/P_hrdB_scnR1 at 2 and 5 days of fermentation, while transcription of *scnR1* was notably reduced in the NEAU6/P_hrdB_gvmR2 strain at day 5 (Figure 5B), indicating that ScnR1 and GvmR2 can engage in reciprocal transcriptional repression.

## 4. Discussion

As a natural product pesticide produced by *Streptomyces* species, guvermectin biosynthesis is tightly regulated by a complex hierarchical regulatory network that includes cluster-situated and pleiotropic/global regulators [23,24,25]. While previous work has identified two proximal LacI-family regulators, GvmR (an activator) and GvmR2 (a repressor), as important players in controlling guvermectin production [8], the roles of distal TRs, located far from the guvermectin BGC, remain largely unexplored. Here, we identified *scnR1*, a highly conserved LacI-family TR located distantly from the guvermectin BGC, and demonstrated its role as a competitive repressor of guvermectin biosynthesis. ScnR1 binds to the same promoter regions as GvmR, specifically the promoters of *gvmR*, *gvmA*, and *O1*, thereby antagonizing the transcriptional activation mediated by GvmR (Figure 6). Furthermore, we uncovered a reciprocal feedback inhibition mechanism between ScnR1 and the cluster-adjacent repressor GvmR2, revealing a multi-layered regulatory architecture controlling guvermectin BGC expression (Figure 6). These results revealed regulatory patterns, including competitive DNA binding at shared promoters and cross-feedback control, that orchestrate guvermectin biosynthesis, advancing our understanding of the multi-level regulatory network that governs the synthesis of NPs in *Streptomyces*.

An increasing number of studies have demonstrated that CSRs are not only capable of activating or repressing the expression of the BGCs they reside in, but can also regulate the activities of other NP BGCs in the genome. For instance, in *Streptomyces venezuelae*, the CSR-type activator JadR1 directly represses the chloramphenicol BGC [26]; whereas in *Streptomyces avermitilis*, the CSR-type activator AveR activates the avermectin BGC while simultaneously repressing the oligomycin BGC [27]. These findings indicate that CSRs exhibit pleiotropic regulatory behaviors and can control the biosynthesis of multiple NPs within the same species. Additionally, some CSRs can exert feedback regulation on their upstream regulators. For example, in *Streptomyces coelicolor*, constitutive overexpression of the activator *redZ* within the actinorhodin BGC significantly enhances the expression of its upper-layer regulator, *afsR2* [23]. However, whether such feedback regulation of upstream regulators by CSRs is a widespread phenomenon remains uncertain. Here, we discovered that the CSR-type activator of the guvermectin BGC, GvmR, directly activates the expression of the extra-cluster regulator *scnR1*. *scnR1* is not located within any NP BGC on the chromosome and is highly conserved across *Streptomyces* species. It inhibits the expression of *gvmR*, thereby repressing guvermectin BGC expression. The feedback regulatory loop between GvmR and ScnR1 suggests that CSRs may precisely regulate NP production by feedback control of hierarchical regulators, which may be an important metabolic control strategy in microorganisms, providing a novel perspective for understanding the intricate regulation of secondary metabolism in *Streptomyces*.

Overexpression of CSR-type activators is a widely used approach for enhancing the production of NPs in *Streptomyces* [14]. However, this approach does not always yield successful results. For example, overexpression of *aveR* from the avermectin BGC and *milR* from the milbemycin BGC resulted in varying degrees of decrease in the production of avermectin and milbemycin, respectively [28,29]. This phenomenon is often attributed to the complex regulatory networks in *Streptomyces*, but the specific mechanisms remain poorly understood. In a previous study, overexpression of two copies of *gvmR* in *S. caniferus* NEAU6 resulted in a decrease in guvermectin production, likely due to the activation-repression feedback loop between GvmR and its inhibitor, GvmR2 [8]. In that study, we circumvented this feedback repression by replacing the native promoter with a strong constitutive promoter for *gvmR*, which successfully led to the high production of guvermectin [8]. In the present study, we identified an additional activation-repression feedback loop between GvmR and ScnR1. Specifically, overexpression of *gvmR* also increased *scnR1* expression, which in turn enhanced the repressive effect of ScnR1 on *gvmR*. This competitive binding to shared target promoters by ScnR1 weakened the promotive effect of GvmR on guvermectin biosynthesis, ultimately hindering the expected increase in guvermectin production. These findings reveal the complex feedback inhibition induced by overexpression of CSR-type activators and offer valuable theoretical guidance for engineering cluster-situated activators aimed at enhancing NP yields.

In this study, ScnR1, GvmR, and GvmR2 are all LacI-family TRs, which regulate each other and can competitively bind to identical promoter targets within the guvermectin BGC, forming a sophisticated, multi-layered local regulatory network that coordinates the synthesis of guvermectin (Figure 6). LacI-family TRs are allosteric DNA-binding regulators [30,31], and the ligands they bind are usually closely related to the metabolic processes they control. The binding of ligands induces conformational changes in the LacI-family TRs, thereby modulating their regulatory activity at target promoters. To explore whether the three TRs bind small molecules, we performed molecular docking to analyze the interactions between ten metabolites associated with guvermectin biosynthesis and the three regulatory proteins [32,33]. The ten metabolites included guvermectin, intermediates like psicofuranine 6′-phosphate and psicofuranine, and biosynthetic precursors like D-glucose, D-fructose 6-phosphate, D-psicose 6-phosphate, PPPP, ATP, AMP, and adenine [3,7]. According to the docking scores, adenine, AMP, psicofuranine, and guvermectin were the top candidates with a potential strong affinity for ScnR1, GvmR, and GvmR2. These compounds were subsequently tested in EMSAs with ScnR1, GvmR, and GvmR2. Surprisingly, the addition of these metabolites failed to alter the DNA binding activities of the three proteins, indicating that their activities may not be regulated by metabolites linked to guvermectin biosynthesis. However, these three regulatory proteins all feature a C-terminal sensor domain capable of binding diverse ligands, including sugars, sugar phosphates, sugar acids, and purines [31]. In our future work, we will expand the ligand library to identify the ligands that these regulatory proteins respond to, and construct a regulatory network containing metabolites (ligands) to more precisely understand the regulatory mechanisms of secondary metabolism in *Streptomyces* represented by guvermectin.

## 5. Conclusions

This work investigated the intricate regulatory mechanisms governing guvermectin production in *S. caniferus* NEAU6, with a focus on the role of ScnR1. Our results demonstrated that ScnR1, an extra-cluster LacI-family regulator, is directly activated by the CSR-type activator GvmR and inhibits guvermectin production by competitive binding with GvmR at shared promoter regions (the bidirectional *gvmR*-*gvmA* promoter and the *O1* promoter). Additionally, a reciprocal feedback loop between ScnR1 and the cluster-adjacent repressor GvmR2 was revealed, adding another layer of complexity to the regulatory network controlling guvermectin biosynthesis. This multi-layered regulatory network, involving competitive binding and feedback mechanisms, offers new insights into the fine-tuned control of natural product biosynthesis in *Streptomyces*.

## Figures and Tables

**Figure 1 biology-14-00813-f001:**
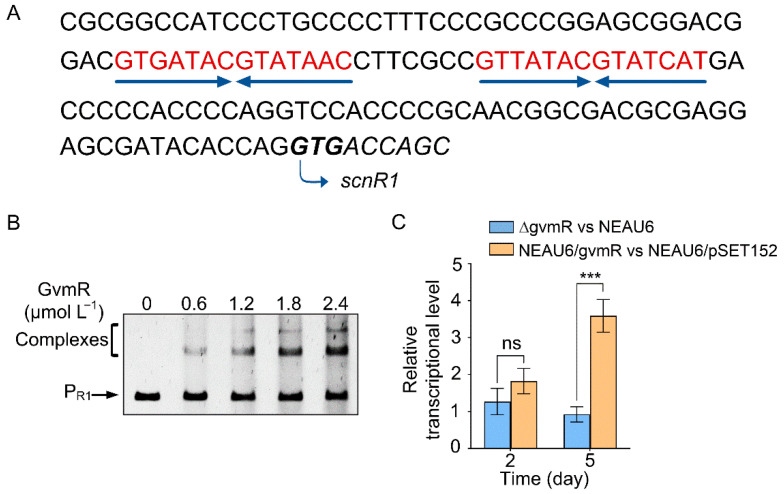
Identification of *scnR1* as a direct regulatory target of GvmR. (**A**) Sequence analysis of the *scnR1* promoter region. The translation start site is indicated by a curved arrow. Palindromic sequences are shown in red and marked with dark blue arrows. (**B**) EMSAs to assess GvmR binding to the *scnR1* promoter region (P_R1_). (**C**) Effects of *gvmR* inactivation and overexpression on *scnR1* transcription (normalized to expression in NEAU6 or NEAU6/pSET152 at days 2 and 5, and to the 16S rRNA gene as the internal control). ns, not significant; *** *p* < 0.001.

**Figure 2 biology-14-00813-f002:**
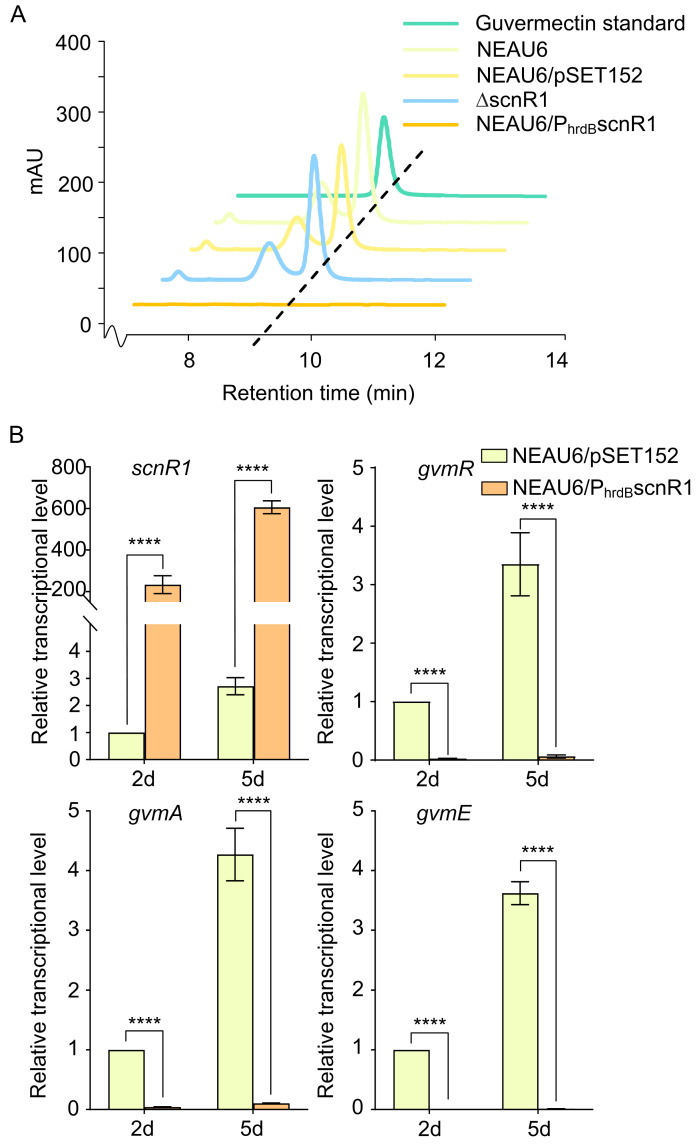
Effects of *scnR1* on guvermectin production and guvermectin BGC expression. (**A**) HPLC analysis of guvermectin production in NEAU6, NEAU6/pSET152, NEAU6/P_hrdB_scnR1, and ΔscnR1. (**B**) Transcriptional analysis of *scnR1*, *gvmR*, *gvmA*, and *gvmE* in NEAU6/pSET152 and NEAU6/P_hrdB_scnR1. Gene expression in NEAU6/P_hrdB_scnR1 was normalized to expression levels in NEAU6/pSET152 at each time point. **** *p* < 0.0001.

**Figure 3 biology-14-00813-f003:**
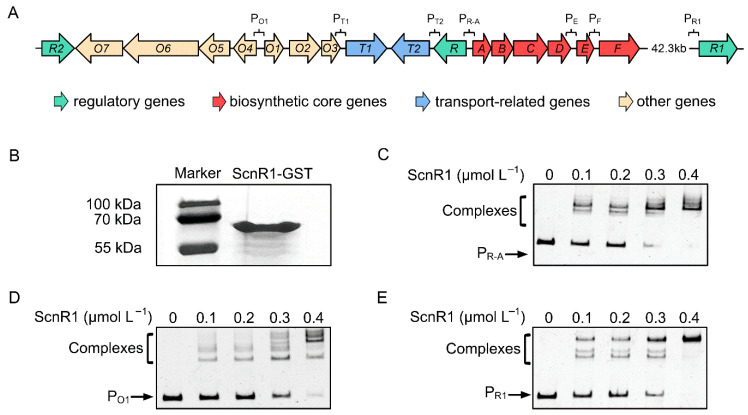
EMSAs demonstrating the binding of ScnR1 to its target promoters. (**A**) A schematic representation of the promoter regions within the guvermectin BGC, and the promoter region of *scnR1*. (**B**) SDS-PAGE profile of purified GST-tagged ScnR1. (**C**) EMSA showing ScnR1 binding to the bidirectional *gvmR*–*gvmA* promoter region. (**D**) EMSA depicting ScnR1 interaction with the *O1* promoter region. (**E**) EMSA showing ScnR1 interaction with its own promoter region.

**Figure 4 biology-14-00813-f004:**
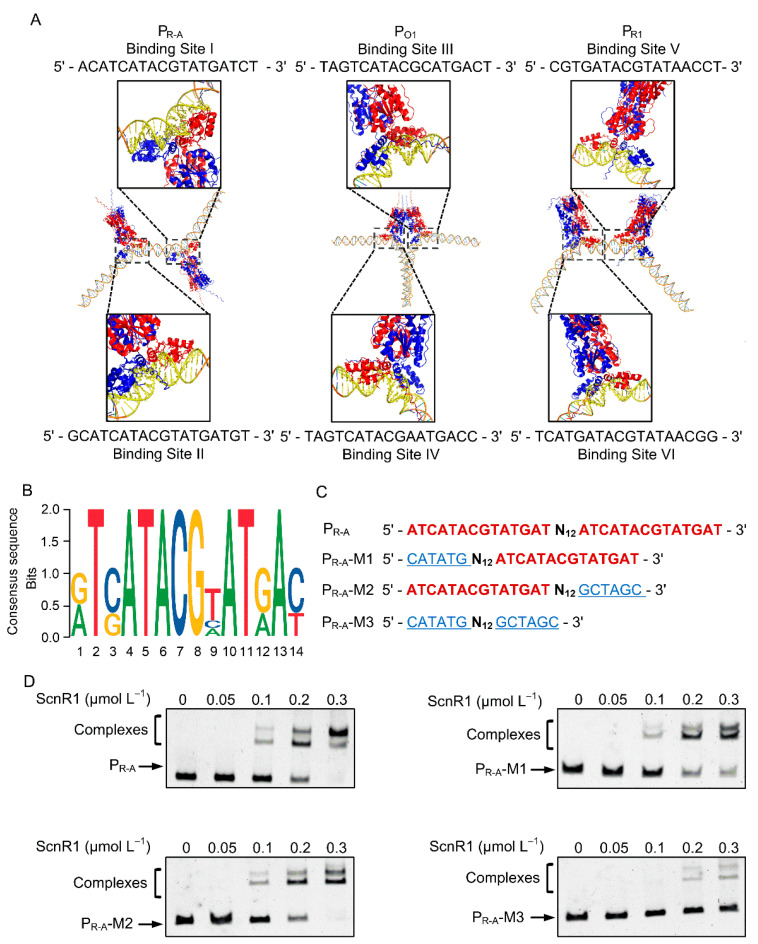
Determination of ScnR1 conserved binding sites. (**A**) Prediction of ScnR1 binding sites using AlphaFold 3. (**B**) Analysis of the conserved sequences within these binding sites. (**C**) Diagrams of P_R-A_ palindromic sequences with introduced mutations (blue highlights). (**D**) EMSA results for ScnR1 binding to P_R-A_ and its mutants.

**Figure 5 biology-14-00813-f005:**
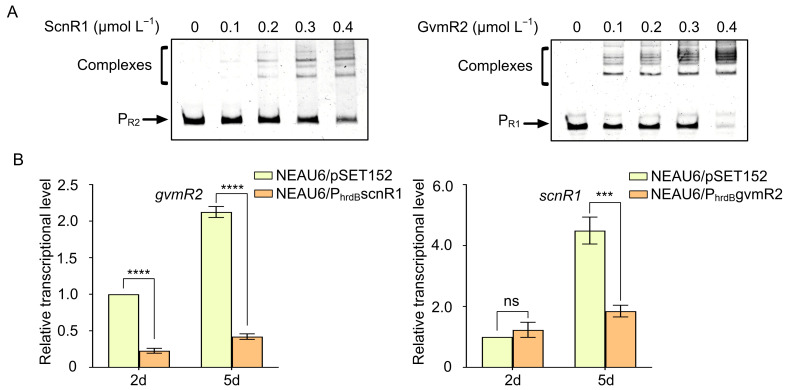
Regulatory relationships between *scnR1* and *gvmR2*. (**A**) EMSAs showing the interaction of ScnR1 and GvmR2 with the promoter regions of each other’s encoding genes. (**B**) qRT-PCR analysis to assess the impact of overexpression of *scnR1* and *gvmR2* on each other’s expression. ns, not significant; *** *p* < 0.001, **** *p* < 0.0001.

**Figure 6 biology-14-00813-f006:**
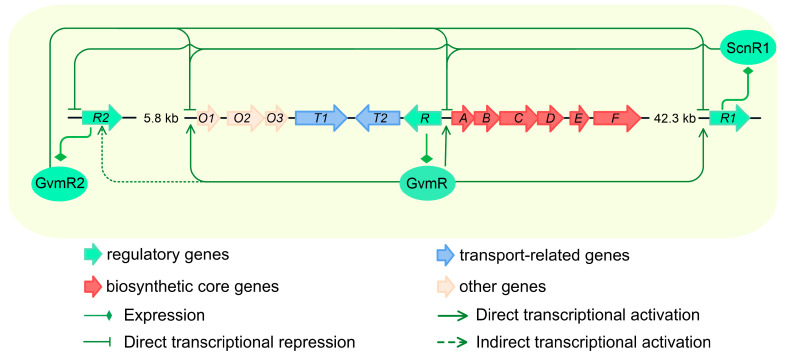
ScnR1-mediated dual regulatory mechanisms in guvermectin biosynthesis. ScnR1 is a key repressor in the biosynthesis of guvermectin. It controls guvermectin BGC expression through a dual-mode regulatory mechanism. First, ScnR1 directly binds to the *gvmR-gvmA* bidirectional promoter region and the *O1* promoter region, where it competitively inhibits the transcriptional activation of guvermectin BGC mediated by the cluster-situated activator GvmR through steric hindrance. Second, ScnR1 interacts with GvmR and the guvermectin BGC-adjacent repressor GvmR2, respectively, forming cross-regulatory circuits. The specific regulatory interactions are as follows: (i) ScnR1 directly binds to the *gvmR* promoter to inhibit its expression, while GvmR reciprocally activates *scnR1* expression by binding to its promoter, forming a repression-activation feedback loop; (ii) ScnR1 and GvmR2 form a mutually inhibitory regulatory module, in which ScnR1 represses *gvmR2* expression, and GvmR2, in turn, can bind to the *scnR1* promoter and repress its expression. These competitive binding interactions and reciprocal feedback regulatory behaviors among the three LacI-family transcriptional regulators indicate the intricate and interdependent transcriptional regulation involved in guvermectin biosynthesis.

## Data Availability

This manuscript builds upon the preprint published by the same authors on bioRxiv (bioRxiv: https://www.biorxiv.org/content/10.1101/2023.11.19.566996v1, accessed on 2 July 2025). Figure 1, Figure 2, Figure 3D,E and Figure 4D are derived from the preprint, with some panels optimized for clarity and additional data included to strengthen the findings. Specifically, we have refined the visualization of qPCR results in Figure 2B, included the effect of *gvmR* overexpression on *scnR1* expression in Figure 1C, and added *scnR1* deletion analysis on guvermectin production in Figure 2A. Figure 3A–C, Figure 4A–C, Figure 5 and Figure 6 are new additions in this manuscript, presenting additional experiments and results. Critically, this work presents new findings, particularly the exploration of the interactions and regulatory relationships between ScnR1 and other key regulators, GvmR and GvmR2.

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
