# Peer review of "ScnR1-Mediated Competitive DNA Binding and Feedback Inhibition Regulate Guvermectin Biosynthesis in *Streptomyces caniferus"

_biology, 2025, doi:10.3390/biology14070813_

Round 1

Reviewer 1 Report

Comments and Suggestions for Authors

The manuscript describes a novel repressor that regulates the expression of a biosynthetic gene cluster responsible for guvermectin production in Streptomyces caniferus. The presented results are of high importance, as they contribute to a better understanding of the transcriptional regulation of gene clusters involved in the synthesis of bioactive compounds in bacteria of this genus.

Overall, the study is well thought out; the results are clearly presented and well explained. However, I have some suggestions:

  • The plasmid pGRK should be described in more detail, including its features, either in the Methods section or in Supplementary table S1.
  • The description of the EMSA method is too limited. It is stated that EMSA was conducted as described by Wei et al., 2014. However, that article specifies which primers were radiolabeled, and that PCR products—either alone or in complexes with protein—were visualized after polyacrylamide gel electrophoresis by autoradiography. Was the same approach used in this study?
  • The meaning of the abbreviation "CSR" is not provided in the text.
  • It is stated that the mutants were confirmed not only by PCR but also by sequencing. The sequencing results should be included in the supplementary material.

Author Response

  1. Comments and Suggestions for Authors

The manuscript describes a novel repressor that regulates the expression of a biosynthetic gene cluster responsible for guvermectin production in Streptomyces caniferus. The presented results are of high importance, as they contribute to a better understanding of the transcriptional regulation of gene clusters involved in the synthesis of bioactive compounds in bacteria of this genus.

Response:

Thank you for your careful examination and positive evaluation of our manuscript. We have made revisions carefully based on your comments. The detailed responses are provided below.

Overall, the study is well thought out; the results are clearly presented and well explained. However, I have some suggestions:

  • The plasmid pGRK should be described in more detail, including its features, either in the Methods section or in Supplementary table S1.

Response:

Thank you for your comment. However, we believe there might have been some confusion regarding the plasmid mentioned. The plasmid pGRK is not included in our study. In our manuscript, we described the construction of the following plasmids: pSET152::PhrdBscnR1 for overexpression of scnR1, pKC1139::scnR1 for deletion of scnR1, and pGEX-4T-1::scnR1 for the protein expression of scnR1. The construction methods for the three plasmids are detailed in the Materials and Methods section. Please refer to lines 118-123, lines 127-134, and lines 155-158.

  • The description of the EMSA method is too limited. It is stated that EMSA was conducted as described by Wei et al., 2014. However, that article specifies which primers were radiolabeled, and that PCR products—either alone or in complexes with protein—were visualized after polyacrylamide gel electrophoresis by autoradiography. Was the same approach used in this study?

Response: Thank you for your careful examination of the experimental details. In the EMSA experiment, we referred to the method described by Wei et al. but made some modifications to the specific experimental procedure. In this study, we did not use radioactively labeled primers for PCR amplification of promoter probes. Instead, we employed SYBR Gold Nucleic Acid Gel Stain for fluorescence staining and used a gel imaging system for detection, rather than autoradiography. We have made appropriate revisions and additional clarifications in the EMSA method section of the manuscript to ensure that the method description is more accurate (lines 166-175).

  • The meaning of the abbreviation "CSR" is not provided in the text.

Response: Thank you for your correction. The abbreviation “CSR” refers to “cluster-situated regulator”. We have provided the meaning of “CSR” in the text as suggested (line 71).

  • It is stated that the mutants were confirmed not only by PCR but also by sequencing. The sequencing results should be included in the supplementary material.

Response: Thank you for your suggestion. The sequencing results have been included in the supplementary material Figure S1B.

Reviewer 2 Report

Comments and Suggestions for Authors

Guvermectin has broad applications in agriculture, making it important to understand the regulatory mechanisms of its biosynthesis for enhanced production. In this paper, Shi et al. report the identification of a new transcriptional regulator, ScnR1, which acts as a repressor of guvermectin biosynthesis in Streptomyces. The study is thoughtfully designed and includes substantial experimental work. However, there are still concerns regarding data interpretation that should be addressed before publication.

  1. In Figure 4D, the authors claimed that the single-site mutation markedly weakens the binding of ScnR1 to the promoter at higher concentration by comparing the upper band of the complex. However, the 2nd complex band was stronger for the promoters with mutations, and there is no clear difference in signal between the unbound PR-A and PR-A-M2. These observations suggest that the M1 region may be more critical for binding than the authors have emphasized. I recommend a more detailed reinterpretation of the EMSA results, ideally supported by quantification of all band intensities using appropriate imaging software.
  2. Figure 5B is missing a y-axis label. Please revise the figure accordingly.

Author Response

  1. Comments and Suggestions for Authors

Guvermectin has broad applications in agriculture, making it important to understand the regulatory mechanisms of its biosynthesis for enhanced production. In this paper, Shi et al. report the identification of a new transcriptional regulator, ScnR1, which acts as a repressor of guvermectin biosynthesis in Streptomyces. The study is thoughtfully designed and includes substantial experimental work. However, there are still concerns regarding data interpretation that should be addressed before publication.

Response:

Thank you for your careful examination, positive evaluation, and constructive suggestions on our manuscript. We have made revisions carefully based on your comments. The detailed responses are provided below.

  1. In Figure 4D, the authors claimed that the single-site mutation markedly weakens the binding of ScnR1 to the promoter at higher concentration by comparing the upper band of the complex. However, the 2ndcomplex band was stronger for the promoters with mutations, and there is no clear difference in signal between the unbound PR-A and PR-A-M2. These observations suggest that the M1 region may be more critical for binding than the authors have emphasized. I recommend a more detailed reinterpretation of the EMSA results, ideally supported by quantification of all band intensities using appropriate imaging software.

Response:

Thank you for your valuable suggestion. We have revised the interpretation of the EMSA results (lines 304 to 312) to provide a more accurate analysis. Additionally, we have included detailed grayscale ratios of all EMSA bands (free probe, lower complex, and upper complex) in Supplementary Table S5 as suggested.

  1. Figure 5B is missing a y-axis label. Please revise the figure accordingly.

Response:

Thank you for your correction. We revised Figure 5B as suggested. Please refer to the revised Figure 5B.

Reviewer 3 Report

Comments and Suggestions for Authors

The manuscript “ScnR1-mediated competitive DNA binding and feedback inhibition regulate guvermectin biosynthesis in Streptomyces caniferus” is devoted to detailed regulatory mechanisms of natural product biosynthesis involving pleiotropic regulators. The manuscript is well-written, nicely illustrated and methodologically sound. Conclusions are clearly supported by the data presented. This study holds significant interest for microbiologists and provides valuable practical insights into optimizing natural product biosynthesis.

Minor comments:

  1. Given that guvermectin production is central to the study, please provide a complete description of the analytical quantification procedure, including HPLC conditions, columns, standards, and details regarding the preparation and calibration curves.
  2. Figure 2A currently does not clearly illustrate differences in guvermectin production levels. A clear comparison of retention times and peak areas, preferably presented in an accompanying supplementary table, would enhance interpretation.
  3. Discussion section would benefit from quantitative assessment of guvermectin production achievable with the newly characterized regulatory complex. Comparison of productivity with strains currently used for guvermectin production would further highlight the practical implications and significance of this research.

Author Response

  1. Comments and Suggestions for Authors

The manuscript “ScnR1-mediated competitive DNA binding and feedback inhibition regulate guvermectin biosynthesis in Streptomyces caniferus” is devoted to detailed regulatory mechanisms of natural product biosynthesis involving pleiotropic regulators. The manuscript is well-written, nicely illustrated and methodologically sound. Conclusions are clearly supported by the data presented. This study holds significant interest for microbiologists and provides valuable practical insights into optimizing natural product biosynthesis.

Response:

Thank you for your positive evaluation of our work.

Minor comments:

  1. Given that guvermectin production is central to the study, please provide a complete description of the analytical quantification procedure, including HPLC conditions, columns, standards, and details regarding the preparation and calibration curves.

Response:

Thank you for your suggestion. We have added relevant information to the Materials and Methods section. Please refer to lines 139 to 148.

Figure 2A currently does not clearly illustrate differences in guvermectin production levels. A clear comparison of retention times and peak areas, preferably presented in an accompanying supplementary table, would enhance interpretation.

Response:

Thank you for your suggestion. We have provided detailed retention times and peak areas in supplementary Table S4 as suggested.

  1. Discussion section would benefit from quantitative assessment of guvermectin production achievable with the newly characterized regulatory complex. Comparison of productivity with strains currently used for guvermectin production would further highlight the practical implications and significance of this research.

Response:

Thank you for your suggestion. We have added Table S4, which provides detailed data on the effects of scnR1 overexpression and deletion on guvermectin production. To ensure the accuracy of the quantitative assessment of guvermectin production in each strain, we replaced the chromatograms in the original Figure 2A, which were obtained earlier and lacked standard concentration information, with newly obtained chromatograms from the same strains under identical detection conditions. The concentration of the standard used in the new chromatograms is clearly recorded. The updated chromatograms are consistent with the original conclusions of the manuscript. This revision was made mainly to improve the accuracy of guvermectin titer determination for each strain and does not impact the key conclusions of the study.

In this study, we identified a novel repressor, ScnR1, that negatively regulates guvermectin biosynthesis. Overexpression of scnR1 completely abolished guvermectin production, while its deletion mutant showed no significant impact on guvermectin production in the current fermentation medium used for production. Although genetic manipulation of scnR1 did not significantly increase guvermectin production, the competitive binding and feedback inhibition mechanisms mediated by ScnR1, and its close interplay with GvmR and GvmR2, explained why simple overexpression of cluster-situated activators fails to increase yields: excess activator expression triggers feedback inhibition from upstream transcriptional regulators, limiting further yield enhancement. These findings provide valuable theoretical guidance for rationally engineering regulatory networks aimed at enhancing NP yields. For related discussions please refer to lines 390 to 408.

Round 2

Reviewer 2 Report

Comments and Suggestions for Authors

Thank you for addressing my concerns. I now fully support its acceptance.